# Decentralized Federated Learning with Function Space Regularization

## Abstract

In this work we propose FedFun, a novel framework for decentralized federated learning that enforces consensus across clients in function space rather than parameter space. By framing agreement as a regularization penalty in a Hilbert space of hypotheses, our method allows optimization using proximal gradient updates that encourage similarity between neighboring models while supporting both parametric and non-parametric learners. This function space perspective yields theoretical advantages, including broad convergence guarantees even when individual client objectives are non-convex in parameter space, and improved robustness to client heterogeneity. We provide convergence analysis under mild assumptions, demonstrate compatibility with models like neural networks and decision trees, and empirically evaluate implementations of FedFun on various sample datasets.

## 1 Introduction

Federated Learning (FL) trains models across a network of devices or silos, called clients, which may include smartphones, edge servers, IoT devices, or institutional data centers. Unlike traditional centralized approaches, where data are collected and processed on a central server, FL assumes that the data must remain on the clients without being transmitted or shared. This addresses a wide range of real-world limitations that would otherwise prevent the application of machine learning or render it difficult, such as prohibitions on data sharing, privacy concerns, or data storage and transfer limitations.

Broadly speaking, FL methods are either centralized or decentralized. Centralized methods make use of a central server that does not need to contribute any data but aggregates information from clients, often in the form of model updates, to coordinate the learning process. For example, the most prominent FL algorithm, Federated Averaging (FedAvg) from McMahan et al. (2017), averages the weights of client models on the central server and then sends the average model back to the clients to continue training. Centralized approaches are simple, relatively easy to implement and analyze, and widely used. However, relying on a central server can be detrimental in some situations. It is a bottleneck for communication and computation; as a single point of failure, it is a vulnerability in the system; and it may be impractical or infeasible in cases involving unreliable networks, ad-hoc field communications, or particular privacy concerns.

In contrast, Decentralized Federated Learning (DFL) methods, such as decentralized federated averaging (DFedAvg) introduced by Sun et al. (2022), do not depend on a central server and instead rely on peer-to-peer communication. Such approaches can alleviate the bottleneck, improve robustness to system failures, and improve practicality in resource-limited settings. Comparison and optimization of various frameworks for decentralized FL, each with unique features and trade-offs involving privacy, fairness, convergence rates, and robustness, is an ongoing area of research (Beltrán et al., 2023).

**Contributions:** In this work we lay the foundations for a new approach to federated learning, FedFun, that views the optimization and enforcement of consensus of client models in function space. In particular, our approach is an iterative process in which clients exchange models with their neighbors and then learn a model on their data with Function Space Regularization (FSR), which penalizes disagreement between client and neighboring models. This update can be analyzed as a proximal gradient method, a popular algorithm for convex optimization. The regularization requires computing inner products and norms of functions, which

is expensive in general. However, where it can be efficiently applied, it unlocks a few advantages and useful capabilities not available when optimizing from a strictly parametric perspective:

- Broadly applicable theoretical convergence. In parametric machine learning, the objective function is commonly non-convex in the parameters, making convergence analysis challenging. However, the analogous learning objective is often convex in function space, guaranteeing convergence of the federated learning iteration so long as the local learning problems are solved near-optimally.

- Support for non-parametric models. Non-parametric models may offer advantages such as lower burden of architecture design and hyperparameter choice, efficient optimization with no special hardware requirements, small memory footprint, or reliable performance on certain problems. Existing FL methods often aggregate models or enforce agreement by some operation on their parameters, and are therefore incompatible with non-parametric models.

- Robustness to client heterogeneity. A key limitation of many FL algorithms is their vulnerability to client heterogeneity (Zhang et al., 2021; Wen et al., 2023; Beltrán et al., 2023; Ye et al., 2023), in particular due to differences in model designs or local data distributions. The negative effect of the latter on FedAvg convergence is well studied both theoretically and empirically, and is sensitive to longer local training between communication rounds (Li et al., 2019; Wang et al., 2019; Li et al., 2020; Karimireddy et al., 2020; Dhawan et al., 2024). In addition to supporting a broad class of model designs, our function space approach is relatively robust to data heterogeneity, requiring relatively few communication rounds and performing even when clients have data split by class.

## 2 Related Work

Beltrán et al. (2023) offer a recent review of decentralized FL, including an in-depth comparison between centralized and decentralized approaches. Among others, they highlight client data heterogeneity and efficiency of computation, storage, and communication as open challenges. The prominent FedAvg algorithm has a decentralized variant called decentralized federated averaging with momentum (DFedAvgM) that removes the need for a centralized server (Sun et al., 2022). Each client performs a fixed number of gradient updates with momentum, then broadcasts its model. Clients then average the model parameters they receive from their neighbors and begin a new round of gradient updates.

Similarly to ours, other prior methods are based on a regularization that penalizes disagreement between neighboring models. Vanhaesebrouck et al. (2017) propose a method of asynchronous model propagating and updates with convergence in expectation to the optimal solution for convex quadratic objectives. For more general convex objectives, they propose a similar method based on the alternating-direction method of multipliers (ADMM) (Boyd et al., 2011), a popular method for distributed convex optimization. This approach additionally communicates a set of dual variables with the same structure as the model parameters. Almeida & Xavier (2018) propose the decentralized Jacobi asynchronous method (DJAM), which similarly has clients asynchronously update and communicate models, but is based on a block-coordinate descent paradigm. They obtain convergence with probability 1 for strongly convex objectives and empirically demonstrate a similar convergence rate to the ADMM variant in Vanhaesebrouck et al. (2017) without the need to tune a hyperparameter.

These methods are presented as learning personalized models since the regularization encourages similarity of neighbor models, but does not require exact consensus, with the tradeoff of agreement and personalization determined by a hyperparameter. We employ our approach similarly, although we also show how increasing regularization yields convergence to the optimal consensus model. Other approaches for personalized federated learning, such as Arivazhagan et al. (2019), T Dinh et al. (2020), and Long et al. (2024), employ strategies for learning that aim to improve local performance by decoupling personal and global models.

A popular centralized algorithm FedProx (Li et al., 2020), offers sublinear convergence to a stationary point of the FL objective assuming bounds on client dissimilarity and milder notions than risk convexity. Work in Yuan & Li (2022) extends these results by showing that one can reduce assumptions on the risk and obtain FedProx convergence independent of dissimilarity, at the cost of a slower convergence rate. In line with other

results above, Hanzely & Richtárik (2020) consider a relaxed problem similar to FedProx and obtain linear convergence results at the cost of strong convexity of local risks.

Among penalty-based methods, the most similar to ours, Bastianello & Dall'Anese (2021) uses the distributed proximal gradient method (DPGM) for synchronous distributed optimization of the sum of strongly convex smooth functions and possibly non-smooth convex functions of the scalar input. They frame the penalized objective as a relaxation of the consensus-constrained objective and show convergence to a neighborhood of the constrained optimum, including for an inexact variant where the updates are assumed to be subject to error with limited expected magnitude.

If our method is used in parameter space instead of function space, it is much like a synchronous variant of DJAM, or a special case of a multidimensional DPGM. We make similar assumptions to Bastianello & Dall'Anese (2021), although we require quadratic rather than strongly convex risks, and obtain similar linear convergence guarantees; however, ours is applicable to a much wider range of realistic learning problems since learning objectives may be convex in function space, even if they are non-convex in parameter space.

Note that aforementioned work all relies on a parametric perspective, in contrast to our function space approach. This approach (outside the federated learning setting) is suggested by Benjamin et al. (2019), who empirically explore a method for FSR, motivated by observed differences in optimization trajectories when viewed in function vs. parameter space. Subsequently, Dhawan et al. (2024) explore the benefits of applying this approach to federated learning, empirically showcasing the skill of a centralized algorithm FedFish that implements FSR using a parametric approximation of KL divergence. They cite advantages including robustness to long local training times, where FedAvg is known to struggle. Our algorithm also provides this benefit, allowing clients to fully optimize models before requiring communication.

In parallel, other related approaches motivated by model heterogeneity rely on knowledge distillation (Ye et al., 2023). Common of these approaches, Li & Wang (2019) explore a centralized approach where clients iteratively do local learning on private data, followed by knowledge distillation using the average of client models on some public data, computed by a central server. Fang & Ye (2022) learn client models for image classification iteratively doing decentralized local learning with knowledge distillation and client confidence scores. Perhaps the closest in this line of work to our approach, Lin et al. (2020) demonstrate a centralized algorithm FedDF iteratively applying local learning and distillation without relying on shared public data. This is the only approach we are aware of with a convergence analysis, which considers binary classification with a specified loss function, and bounds the risk attainable by the consensus in terms of the risk attainable learning from the global data distribution. While the interlaced local learning and distillation steps in these approaches differ from the FSR formulation in FedFun, it is worth highlighting that both strategies are designed for non-parametric model alignment: For positive densities the $L^2$ distance, used in both Benjamin et al. (2019) and our implementations, and KL-divergence, used in knowledge distillation and FedFish, just amount to different ways of quantifying the distance between distributions.

## 3 Method

We first formalize our notion of learning viewed in function space, then describe the proposed framework for decentralized federated learning, its convergence properties, and its application to a few model classes.

### 3.1 Proximal Gradient Method in Function Space

A hypothesis $h : \mathcal{X} \to \mathcal{Y}$ is a function that maps input values to a prediction. The goal of a learning algorithm is to select an optimal hypothesis $h^*$ from a hypothesis space $\mathcal{H}$ that minimizes an objective function or risk $R : \mathcal{H} \to \mathbb{R}$. The term *risk* is used to describe an expected loss, usually estimated empirically using the training data. Here, for convenience, we use the term risk and the symbol $R$ to encompass the entire learning objective, including both empirical risk and other components such as regularization.

We assume that the hypothesis space $\mathcal{H}$ is a (real, separable) Hilbert space equipped with an inner product $\langle \cdot, \cdot \rangle_{\mathcal{H}} : \mathcal{H} \times \mathcal{H} \to \mathbb{R}$ and associated norm $\|h\|_{\mathcal{H}}^2 = \langle h, h \rangle_{\mathcal{H}}$. A useful example is the space $L^2(\mathcal{X})$ obtained by considering square-integrable functions from $\mathcal{X} \subseteq \mathbb{R}^p$ to $\mathcal{Y} \subseteq \mathbb{R}^q$, where the commonly associated inner

product is

$$\langle h, g \rangle_{\mathcal{H}} = \int_{\mathcal{X}} \langle h(x), g(x) \rangle_{\mathcal{Y}} \, dx \tag{1}$$

and $\langle \cdot, \cdot \rangle_{\mathcal{Y}}$ is the usual vector inner product on $\mathbb{R}^q$. When $h$ is a model, this inner product may be difficult to compute exactly or even approximate efficiently; we discuss this challenge in Section 5.

This is by no means the only possible Hilbert space or inner product, but it is simple and practical for machine learning applications. Although most model classes do not exactly form a Hilbert space, many are reasonable to analyze as such. For instance, neural networks and decision trees have a *universal approximator* property; informally, sufficiently large models in these families can approximate any function in $L^2$ with arbitrary precision. Other classes, such as linear models, define pragmatically useful subspaces of $L^2$.

In a decentralized federated learning setting, each client $i \in [n]$ has a sample of training data that define its local risk $R_i$, which it can use to learn a local model $h_i$. Each client is able to communicate with some subset of other clients, defining a (symmetric) communication graph where edges between clients indicate the ability to exchange information.

As a starting point, consider consensus learning where the goal is to select a consensus model $h^*$ that minimizes the aggregate risk

$$\bar{R}[h] = \sum_i R_i[h]. \tag{2}$$

It is assumed that the clients may not communicate data, so each risk $R_i$ may only be evaluated at client $i$, and clients communicate by exchanging models. Thus we write an equivalent optimization problem in terms of local models with an agreement constraint:

$$H^* = \arg\min_{\mathbf{h} \in \mathcal{H}^n} \sum_{i \in [n]} R_i[h_i]$$
$$\text{s.t. } h_i = h_j \; \forall i, j \tag{3}$$

where $H^* \subseteq \mathcal{H}^n$ is a nonempty set of optimal consensus models and $\mathbf{h}^* = (h^*, \ldots, h^*) \in H^*$.

Let $L \in \mathbb{R}^{n \times n}$ be a symmetric Laplacian of the communication graph with $L_{i,j} = -1$ if $i$ and $j$ are clients that may directly communicate and $L_{i,i} = -\sum_{j \neq i} L_{i,j}$. To facilitate optimization and enable some degree of personalization to client-specific data, we relax the agreement constraint in (3) to a disagreement penalty $\frac{1}{2}\lambda \sum_{i \neq j} -L_{i,j}\|h_i - h_j\|^2$ for some penalty coefficient $\lambda > 0$. The relaxed optimization problem can then be written:

$$\tilde{H} = \arg\min_{\mathbf{h} \in \mathcal{H}^n} \sum_{i \in [n]} R_i[h_i] + \frac{1}{2}\lambda \langle \mathbf{h}, \mathbf{L}\mathbf{h} \rangle_{\mathcal{H}^n} \tag{4}$$

where $\mathbf{L}$ is a positive operator $(\mathbf{L}\mathbf{h})_i = \sum_j L_{ij} h_j$ on $\mathcal{H}^n$, which is itself a Hilbert space with inner product $\langle \mathbf{h}, \mathbf{g} \rangle_{\mathcal{H}^n} = \sum_i \langle h_i, g_i \rangle_{\mathcal{H}}$, and $\tilde{H}$ is the solution set.

Problems where this formulation would be suitable arise for example in healthcare, where clients may aim to learn the effectiveness of certain treatments but regulatory restrictions limit with whom they can communicate. Similar problems arise in predictive maintenance, for example detecting faults in equipment. In other settings a distributed network of sensors may aim to learn various models e.g. for object detection, but may only be able to communicate with other sensors within a given range.

To solve the problem, we use an iterative process initialized by each client minimizing its local risk: $h_i^{(0)} = \arg\min_h R_i[h]$. Then the clients exchange models with their neighbors on the network and the proximal gradient method, a convex optimization algorithm to minimize the sum of a smooth, differentiable function and a possibly nonsmooth function, yields the separable iterative update:

$$h_i^{k+1} = \arg\min_{h_i \in \mathcal{H}} R_i[h_i] + \frac{1}{2\gamma}\|h_i - h_i^k\|^2 + \lambda \sum_{j \in [n]} L_{i,j} \langle h_i, h_j^k \rangle_{\mathcal{H}} \tag{5}$$

for proximal gradient parameter $\gamma$, which is analogous to a learning rate in generic machine learning algorithms. This depends only on the information available to the client $i$ in iteration $k$ and is solved using a local learning algorithm augmented with a function space regularization.

## 4 Convergence

We analyze the convergence of iteration (5) and the proximity of its solution set $\tilde{H}$ to the consensus solution set $H^*$. We then discuss some practical considerations related to the theoretical convergence. See Appendix A for the proofs omitted here.

**Notation.**

- On a vector in Hilbert space such as $\mathcal{H}$ or $\mathcal{H}^n$, $\|\cdot\|$ denotes the norm induced by the corresponding inner product. On a matrix or linear operator, $\|\cdot\|$ denotes its spectral norm.

- Given $v$ and nonempty set $S$, $d(v, S) = \inf_{s \in S} \|v - s\|$ is the shortest distance from $v$ to $S$. For sets $S$ and $T$, $d(S, T) = \inf_{s \in S, t \in T} \|s - t\|$ is the shortest distance between them.

- $\mathbf{E}$ is the operator such that $(\mathbf{E}h)_i = \frac{1}{n} \sum_j h_j$ for all $i$, projecting $\mathbf{h} \in \mathcal{H}^n$ to consensus.

- Given a linear operator $\mathbf{L}$, $\sigma(\mathbf{L})$ denotes the spectrum of $\mathbf{L}$. We say $\mathbf{L}$ has a spectral gap when there exists $\nu > 0$ s.t. $\sigma(\mathbf{L}) \cap (0, \nu) = \emptyset$.

- Given a linear operator $\mathbf{L}$, we say that $\mathbf{L}$ is positive if $\mathbf{L}$ is positive semidefinite and self-adjoint.

**Assumption 1.** The client risks $R_i$ are convex and have a minimum on $\mathcal{H}$.

Although convexity is a strong assumption in parameter space, virtually all commonly used loss functions are convex in function space, making this broadly applicable.

**Assumption 2.** The communication graph represented by $L$ is connected.

The smallest eigenvalue of a graph Laplacian is always zero. Let $\nu$ be the second-smallest eigenvalue of $L$; $\nu$ is known as the algebraic connectivity of the communication graph and Assumption 2 implies that $\nu > 0$. Moreover, it is straightforward to show that $\sigma(\mathbf{L})$ consists of the eigenvalues of $L$, so $\mathbf{L}$ has a spectral gap and $\|\mathbf{L}\| = \|L\|$.

**Assumption 3.** The proximal gradient parameter $\gamma$ satisfies $0 < \gamma < \frac{1}{\lambda \|L\|}$.

Then the proximal gradient method converges weakly to a solution of the relaxed optimization problem (4) (Combettes & Wajs, 2005, Theorem 3.4 (i)). However, under stronger assumptions, we can show fast convergence to a neighborhood of the solution set of the original constrained optimization problem (3).

**Assumption 4.** For each $i$, $R_i$ is convex quadratic; in particular, there exist positive operators $A_i : \mathcal{H} \to \mathcal{H}$, $a_i \in \mathcal{H}$, and $\alpha_i \in \mathbb{R}$ such that $R_i[h] = \frac{1}{2} \langle h, A_i h \rangle + \langle a_i, h \rangle + \alpha_i$. Moreover, $A_i$ have a spectral gap of at least $\mu > 0$ and commute with each other.

Assumption 4 implies that $R$ is quadratic, in particular $R[\mathbf{h}] = \frac{1}{2} \langle \mathbf{h}, \mathbf{A}\mathbf{h} \rangle + \langle \mathbf{a}, \mathbf{h} \rangle + \alpha$ with (self-adjoint) $\mathbf{A}$ satisfying $(\mathbf{A}\mathbf{h})_i = A_i h_i$, $\mathbf{a}_i = a_i$, and $\alpha = \sum_i \alpha_i$. Thus $R$ is differentiable and $\nabla R$ is Lipschitz continuous with constant $\|\mathbf{A}\|$, and $\sigma(\mathbf{A}) = \bigcup_i \sigma(A_i)$, so $\sigma(\mathbf{A}) \cap (0, \mu) = \emptyset$. The commutativity of $A_i, A_j$ further implies $\sigma(\sum_i A_i) \cap (0, \mu) = \emptyset$. The lemma 1 shows that these gaps in the spectra establish a growth rate of the respective quadratic functions.

**Lemma 1.** *Let $\varphi$ be a quadratic function $\varphi(h) = \frac{1}{2} \langle h, Ah \rangle + \langle a, h \rangle + \alpha$ on a Hilbert space $\mathcal{H}$ with a minimum value $\varphi^*$. If $A$ is positive with spectral gap $\sigma(A) \cap (0, c) = \emptyset$, then*

$$\|\nabla \varphi[h]\| \geq c\, d(h, \arg\min \varphi) \tag{6}$$

$$\varphi[h] \geq \varphi^* + \frac{1}{2} c\, d^2(h, \arg\min \varphi) \tag{7}$$

*for any $h \in \mathcal{H}$.*

Next, Theorem 1 establishes the existence of spectral gap in the operator defining the quadratic objective of the relaxed optimization problem (4).

**Theorem 1.** *There exists some $c > 0$ such that $\sigma(\mathbf{A} + \lambda\mathbf{L}) \cap (0, c) = \emptyset$.*

With these, Theorem 2 establishes linear convergence of iteration (5).

**Theorem 2.** *The distance of the clients' local hypotheses to the relaxed solution set is bounded by*

$$d(\mathbf{h}^{k+1}, \tilde{H}) \leq \frac{1}{\sqrt{1 + \gamma c}} d(\mathbf{h}^k, \tilde{H}). \tag{8}$$

*Proof.* Let $\varphi[\mathbf{h}] = R[\mathbf{h}] + \frac{1}{2}\lambda\langle\mathbf{h}, \mathbf{Lh}\rangle$ denote the minimization objective of the relaxation (4) and $\varphi^* = \min_{\mathbf{h}\in\mathcal{H}^n}\varphi[\mathbf{h}]$. By (Beck & Teboulle, 2009, Lemma 2.3, Remark 2.1), we have the following bound for any $\tilde{\mathbf{h}} \in \tilde{H}$.

$$\varphi[\mathbf{h}^{k+1}] - \varphi^* \leq -\frac{1}{2\gamma}\|\mathbf{h}^{k+1} - \mathbf{h}^k\|^2 - \frac{1}{\gamma}\langle\mathbf{h}^k - \tilde{\mathbf{h}}, \mathbf{h}^{k+1} - \mathbf{h}^k\rangle$$

$$= \frac{1}{2\gamma}\left(\|\mathbf{h}^k - \tilde{\mathbf{h}}\|^2 - \|(\mathbf{h}^{k+1} - \mathbf{h}^k) + (\mathbf{h}^k - \tilde{\mathbf{h}})\|^2\right) = \frac{1}{2\gamma}\left(\|\mathbf{h}^k - \tilde{\mathbf{h}}\|^2 - \|\mathbf{h}^{k+1} - \tilde{\mathbf{h}}\|^2\right).$$

Next Lemma 1 and Theorem 1 imply that $\varphi[\mathbf{h}^{k+1}] \geq \varphi^* + \frac{1}{2}cd^2(\mathbf{h}^{k+1}, \tilde{H})$. Applying this to the above inequality,

$$\frac{1}{2}cd^2(\mathbf{h}^{k+1}, \tilde{H}) \leq \frac{1}{2\gamma}\left(\|\mathbf{h}^k - \tilde{\mathbf{h}}\|^2 - \|\mathbf{h}^{k+1} - \tilde{\mathbf{h}}\|^2\right).$$

Since $d^2(\mathbf{h}^k, \tilde{H}) = \inf_{\mathbf{h}\in\tilde{H}}\|\mathbf{h}^k - \mathbf{h}\|^2$, for any $\epsilon > 0$, there exists $\tilde{\mathbf{h}} \in \tilde{H}$ such that $\|\mathbf{h}^k - \tilde{\mathbf{h}}\|^2 \leq d^2(\mathbf{h}^k, \tilde{H}) + \epsilon$. Then

$$\frac{1}{2}cd^2(\mathbf{h}^{k+1}, \tilde{H}) \leq \frac{1}{2\gamma}\left(d^2(\mathbf{h}^k, \tilde{H}) + \epsilon - d^2(\mathbf{h}^{k+1}, \tilde{H})\right) \text{ for any } \epsilon > 0, \text{ and the claim follows.}$$

$\square$

Now Theorem 3 shows that this solution is within $O(1/\lambda)$ of the consensus optimal solution set $H^*$.

**Theorem 3.** *For a given $\lambda$, for any $\tilde{\mathbf{h}} \in \tilde{H}$,*

$$d(\tilde{\mathbf{h}}, H^*) \leq \frac{\|\mathbf{A}\|}{\lambda\nu}\sqrt{\frac{\|\mathbf{A}\|}{\mu}}\left(1 + \frac{n\|\mathbf{A}\|}{\mu}\right)d(H^*, \arg\min R) \in O(1/\lambda). \tag{9}$$

Together, these theorems tell us that iterations converge quickly to the relaxed optimum and, moreover, that as we increase the penalty coefficient $\lambda$, the relaxed optimum approaches the consensus optimum, that is, the solution to the original problem (3) of finding a globally optimal hypothesis. Further, we lastly note that exact solutions to our subproblems are not required: Suppose update (5) is carried out with additive error $\mathbf{e}^k \in \mathcal{H}^n$ as

$$h_i^{k+1} = e_i^k + \arg\min_{h_i\in\mathcal{H}} R_i[h_i] + \frac{1}{2\gamma}\|h_i - h_i^k\|^2 + \lambda\sum_{j\in[n]} L_{i,j}\langle h_i, h_j^k\rangle. \tag{10}$$

This error can account for imperfect learning algorithms, hypothesis spaces that are not Hilbert spaces but are a $\varepsilon$-cover of one, etc. Incorporating this into Equation (8) via the triangle inequality, we have

$$d(\mathbf{h}^{k+1}, \tilde{H}) \leq \frac{1}{\sqrt{1 + \gamma c}} d(\mathbf{h}^k, \tilde{H}) + \|\mathbf{e}^k\| \tag{11}$$

and, if $\|\mathbf{e}^k\|$ is bounded by a constant for all $k$, then the accumulation of error is bounded by a convergent geometric series. This means that we can expect convergence close to the optimum even if the learning algorithm cannot solve (5) perfectly, which is the case for both neural networks and trees of bounded size.

## 5 Examples

We next discuss how the optimization problem in iteration (5) can be solved by incorporating function space regularization with learning algorithms. We suggest risks satisfying Assumption 4, a model-agnostic strategy using a Monte Carlo method to approximately compute the inner product (1), as well as better model-specific methods for a couple of model classes.

### 5.1 Smoothed Squared Error

While quadratic loss functions are common, the use of a quadratic loss is not sufficient to satisfy Assumption 4 (quadratic risk) because typical point-wise empirical risk cannot be expressed as quadratic using the inner product (1). However, a risk smoothed over a kernel $k : \mathcal{X} \times \mathcal{X} \to \mathbb{R}_{\geq 0}$ combined with a quadratic loss, does satisfy Assumption 4 when the kernel has a minimum non-zero value.

In particular, suppose that each client $i$ has data $\mathbf{x}_j \in \mathbb{R}^p$, $\mathbf{y}_j \in \mathbb{R}^q$, $j \in [N_i]$, and let $k : \mathbb{R}^p \times \mathbb{R}^p$ be a symmetric smoothing kernel that integrates to 1 over the domain. Then define risks as the smoothed sum squared error

$$
\begin{aligned}
R_i[\mathbf{h}_i] &= \sum_j \mathbb{E}_{X_j \sim k(\cdot, \mathbf{x}_j)} \|\mathbf{h}_i(X_j) - \mathbf{y}_j\|^2 \\
&= \sum_j \int_z k(\mathbf{z}, \mathbf{x}_j) \|\mathbf{h}_i(\mathbf{z}) - \mathbf{y}_j\|^2 \, d\mathbf{z} \\
&= \sum_j \int_z k(\mathbf{z}, \mathbf{x}_j)(\mathbf{h}_i(\mathbf{z})^\top \mathbf{h}_i(\mathbf{z}) + \mathbf{y}_j^\top \mathbf{y}_j - 2\mathbf{h}_i(\mathbf{z})^\top \mathbf{y}_j) d\mathbf{z} \\
&= \frac{1}{2} \left\langle \mathbf{h}_i, 2 \sum_j k(\cdot, \mathbf{x}_j)\mathbf{h}_i \right\rangle + \left\langle -2 \sum_j k(\cdot, \mathbf{x}_j)\mathbf{y}_j, \mathbf{h}_i \right\rangle + \sum_j \mathbf{y}_j^\top \mathbf{y}_j
\end{aligned}
$$

where the spectrum of the positive linear operator $\mathbf{h}_i \mapsto \left( \sum_j k(\cdot, \mathbf{x}_j) \right) \mathbf{h}_i(\cdot)$ is the essential range of $\sum_j k(\cdot, \mathbf{x}_j)$. Then $\mu \geq 2 \min_{j, \mathbf{z}|k(\mathbf{z}, \mathbf{x}_j)>0} k(\mathbf{z}, \mathbf{x}_j)$ and $\|\mathbf{A}_i\|$ is bounded by $\|\mathbf{A}_i\| \leq 2 \sum_j \max_{\mathbf{z}} k(\mathbf{z}, \mathbf{x}_j)$; if we assume $k(\cdot, \mathbf{x}_j)$ is the same at each $j$, then bounds for $\mu$ and $\|\mathbf{A}\|$ are simply twice its minimum nonzero value and up to $2 \max_i N_i$ times its maximum respectively, suggesting the use of uniform kernels.

Here, we have intentionally defined the risk as the sum, rather than the mean, of loss values at the training samples so that, when the risks are summed over the clients as in the objective (4), all samples are equally weighted. This also prevents $\mu$ from depending on the total number of samples.

This error smoothing is often important to achieve good performance. Without it, it is possible that, as optimization proceeds, both local risk and disagreement approach zero, but average global accuracy, that is, the accuracy on the union of all training sets, averaged over client models, does not improve. We sometimes observe this in practice when not using error smoothing, especially for moderate to high dimensional data where the coverage of the data over the domain is poor.

### 5.2 Model-agnostic Approximations

The inner product (1) and associated norm can be estimated by a Monte Carlo method arising naturally from the equalities

$$\langle h, g \rangle_{\mathcal{H}} = m(\mathcal{X})\mathbb{E}_{x \sim \mathcal{U}_{\mathcal{X}}}[\langle h(x), g(x) \rangle_{\mathcal{Y}}] \tag{12}$$

$$\|h\|_{\mathcal{H}} = m(\mathcal{X})\mathbb{E}_{x \sim \mathcal{U}_{\mathcal{X}}}[\|h(x)\|_{\mathcal{Y}}] \tag{13}$$

where $\mathcal{U}_{\mathcal{X}}$ is the uniform distribution on $\mathcal{X}$ and $m(\mathcal{X})$ is the Lebesgue measure, or $p$-volume, of $\mathcal{X} \subseteq \mathbb{R}^p$. This, of course, demands that $m(\mathcal{X})$ is finite, but this is of little practical consequence. A similar strategy can be employed for other inner products. If the risk is based on mean squared error, as is recommended by our convergence theory, then this strategy applied to (5) can be reduced to simply incorporating a number

of appropriately weighted random samples from $\mathcal{U}_{\mathcal{X}}$ into the training set with labels the outputs from other models.

Although this is simple and general, it is the least desirable approach overall. The quality of the approximation depends on the number of samples, and as the dimension of $\mathcal{X}$ increases, so does the number of samples required to achieve reasonable coverage. Depending on the type and complexity of the model, it can be expensive to compute the output of several model instances on very many data, and some learning algorithms may not handle very large training sets efficiently. This motivates future work to improve sample efficiency, either by modifying the inner product or using a more efficient sampling strategy.

### 5.3 Neural Networks

Neural networks are typically trained using a minibatch gradient-based optimization algorithm. This motivates a variation of the above model-agnostic concept where new samples are taken at each batch. Recall that we recommend a smoothed squared error loss and defining local risk as the sum, rather than mean, of loss on local data. Then at client $i$, the normalized loss for a batch $(x_1, y_1), \ldots, (x_b, y_b)$ is

$$
\begin{aligned}
&\frac{1}{bq} \sum_{i=1}^{b} \|h(x_i + \epsilon_i) - y_i\|^2 \\
&+ \frac{m(\mathcal{X})}{Nb'q} \sum_{i=1}^{b'} \left[ \frac{1}{2\gamma} \|h(z_i) - h_i^k(z_i)\|^2 + \lambda \sum_{j \neq i} -L_{i,j} \|h(z_i) - h_j^k(z_i)\|^2 \right]
\end{aligned}
\tag{14}
$$

where $b$ is the batch size, $b'$ is the penalty batch size, $N$ is the local training set size, each $\epsilon_i$ is sampled from $k(\cdot, x_i)$, and each $z_i$ is sampled from $\mathcal{U}_{\mathcal{X}}$. For classification, where $\mathcal{Y} = [0,1]^q$, this scaling places the first (risk) term in $[0,1]$ and the second (penalty) term in $[0, \frac{m(\mathcal{X})}{N}(\frac{1}{2\gamma} + \lambda L_{i,i})]$.

### 5.4 Decision Trees

Decision trees are an ideal model class for showcasing the application of this framework. Not only are they non-parametric, making them compatible with function space but not parametric FL methods, but there exists an efficient, exact method for learning trees with both smoothed loss and function space agreement regularization.

For trees, we assume the domain $\mathcal{X}$ is a hyperrectangle, for example, a bounding box of the data. Then a tree can be represented as a collection of nodes $N_i \subseteq \mathcal{X}$ and associated values $v_i \in \mathcal{Y}$. The leaf nodes form a partition of $\mathcal{X}$, and the tree as a function is written

$$
h(x) = \sum_{i \in \text{leaves}(h)} \mathbf{1}\{x \in N_i\} v_i
\tag{15}
$$

and, for a second tree $g$ with nodes $M_j$ and values $w_j$, the inner product (1) is

$$
\langle h, g \rangle = \sum_{i \in \text{leaves}(h)} \sum_{j \in \text{leaves}(g)} m(N_i \cap M_j) \langle v_i, w_j \rangle
\tag{16}
$$

with $m$ the Lebesgue measure, which is easy to compute for the intersection of hyperrectangles. By traversing one tree and tracking the intersecting subtrees of the other, one can avoid computing the zero-measure terms, which are the majority if $h$ and $g$ are similar trees; if $h$ and $g$ are identical, then the number of nonzero terms is just the number of leaves. Since our algorithm penalizes disagreement, the trees are usually similar in practice. However, in the worst possible case, every term may be nonzero, and the cost is proportional to the size of $h$ times the size of $g$.

Next, we need an algorithm for fitting decision trees with smoothed squared error and function space regularization to other trees. Such an algorithm already exists in prior work. Kernel Density Decision Trees (KDDTs) introduced by Good et al. (2022) generalize the classic CART algorithm of Breiman et al. (1984)

for decision tree induction with kernel-smoothed impurity; in particular, with the Gini impurity, KDDT construction minimizes the smoothed squared error we describe in Section 5.1. Splits can be chosen efficiently and optimally as long as the kernel is isotropic, that is, equivalent to the product of its marginal distributions, and those marginal distributions are piecewise-constant. A good example is a box kernel. KDDTs also optionally apply the same smoothing to predictions. For the purpose of function space regularization, we consider the unsmoothed tree, though smoothing can still be used for prediction in practice, and Good et al. (2022) shows that it is usually beneficial to performance. Next, regularization to other trees is accomplished simply by observing that, because each leaf of the other trees contributes squared error uniformly on a hyperrectangle, it can be input directly to the KDDT fitting algorithm as a weighted data point with uniform hyperrectangular kernel and label equal to the corresponding leaf value. Much like the efficient computation of (15), a leaf is tracked only when it intersects the current subtree during construction, so it is efficient when the trees are similar, but can become expensive if they are both large and very dissimilar.

If using unsmoothed squared error with regularization to other trees, tree fitting is more complicated. By an obvious generalization of Theorem 1 of Good et al. (2022), during the search for an optimal split, for each training index $i$ and feature index $f \in [p]$, both $z_f < x_{i,f}$ and $z_f \leq x_{i,f}$, where $z$ is an input to the tree, are candidates for the optimal decision rule. This is achievable with a straightforward alteration of the KDDT fitting algorithm, but it introduces the strange phenomenon of leaves that have zero measure, which are completely absent in the computation of function inner products. Thus, two trees may have zero disagreement, being equivalent in function space, but make completely different predictions on the data. Although this can be circumvented by setting a stopping condition for tree fitting that prevents zero-measure leaves, simply using smoothed error offers a preferably explicit strategy for enforcing continuity around relevant data. Furthermore, regardless of the needs of our federated framework, Good et al. (2022) show that the regularizing effect of smoothed error can greatly increase the performance of single-tree models.

It should also be noted that the KDDT algorithm is only useful when the choice of inner product and smoothing are compatible with its assumptions. If not, we would need a more general fitting algorithm, or lacking that, revert to model-agnostic approximations.

## 6 Experiments

To demonstrate the proposed algorithm, we benchmark our method applied to KDDTs and a small MLP on 12 popular data sets from the UCI Machine Learning Repository (Dua & Graff, 2017) summarized in Table 1 in Appendix B. To compare against the most similar parametric method, we use as a baseline a synchronous variant of DJAM which we call Parameter Space Regularization (PSR). We initially included DFedAvgM as a baseline, but since it trains only for a handful of updates per communication iteration, whereas the other methods fully train a model, it was not able to achieve meaningful performance in the 20 iterations in our experiments, so we omit it from the results. The evaluation is designed as a proof-of-concept; it is left to future work to benchmark the approach against more complex state-of-the-art methods for a variety of learning tasks.

For each data set, we randomly split it into 50% train, 50% test, then split the training data into clients by using k-means to group the features into two clusters. This is a split with high client heterogeneity. We also split the data by class, that is, such that there is one client per class, and each client sees only one class. This is an extreme case of client heterogeneity. To define the communication graph, we sample a random ring (2-regular) graph. Additional experiment details are in Appendix B.

For our FSR methods, we use smoothed squared error with a box kernel with radius $\delta$, that is, $k(\mathbf{z}, \mathbf{x}) \propto \mathbf{1}\{\|\mathbf{z} - \mathbf{x}\|_\infty \leq \delta\}$. Note that after a communication round, each client optimizes starting from their previous local optimum. This often results in faster convergence, presumably by reducing errors accumulated by local solvers. For simplicity, we select $\lambda$ and $\delta$ by training models with a range of values for each and selecting the ones that result in the highest final average global training accuracy.

The results for the cluster-based data split are shown in Figure 1. On most data sets, we see that the FSR-based methods learn faster than PSR, sometimes reaching their best accuracy in just one or two rounds of communication. They also often outperform PSR in final accuracy. The FSR KDDT and FSR

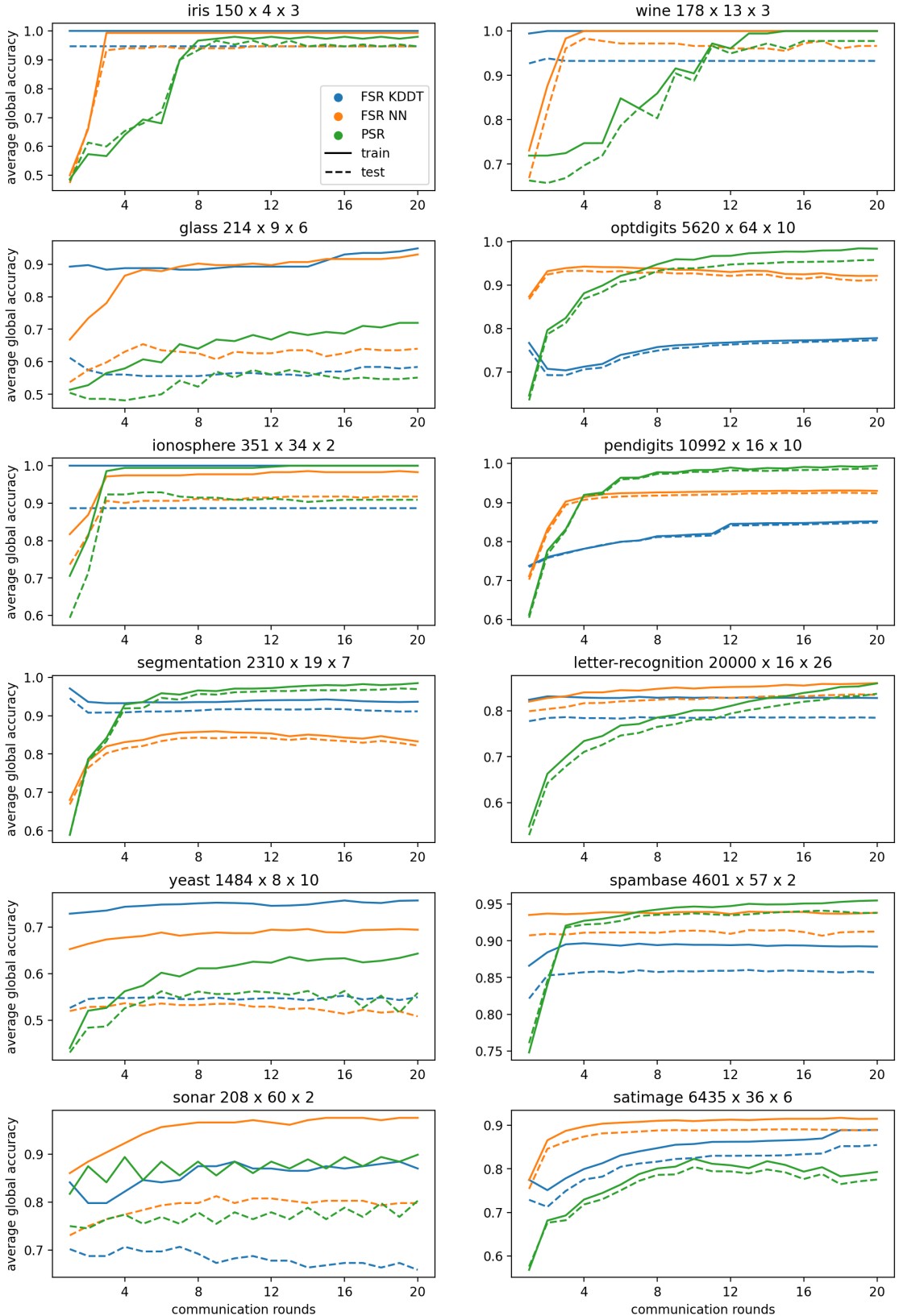

Figure 1: Results for FL experiments with data split by clustering.

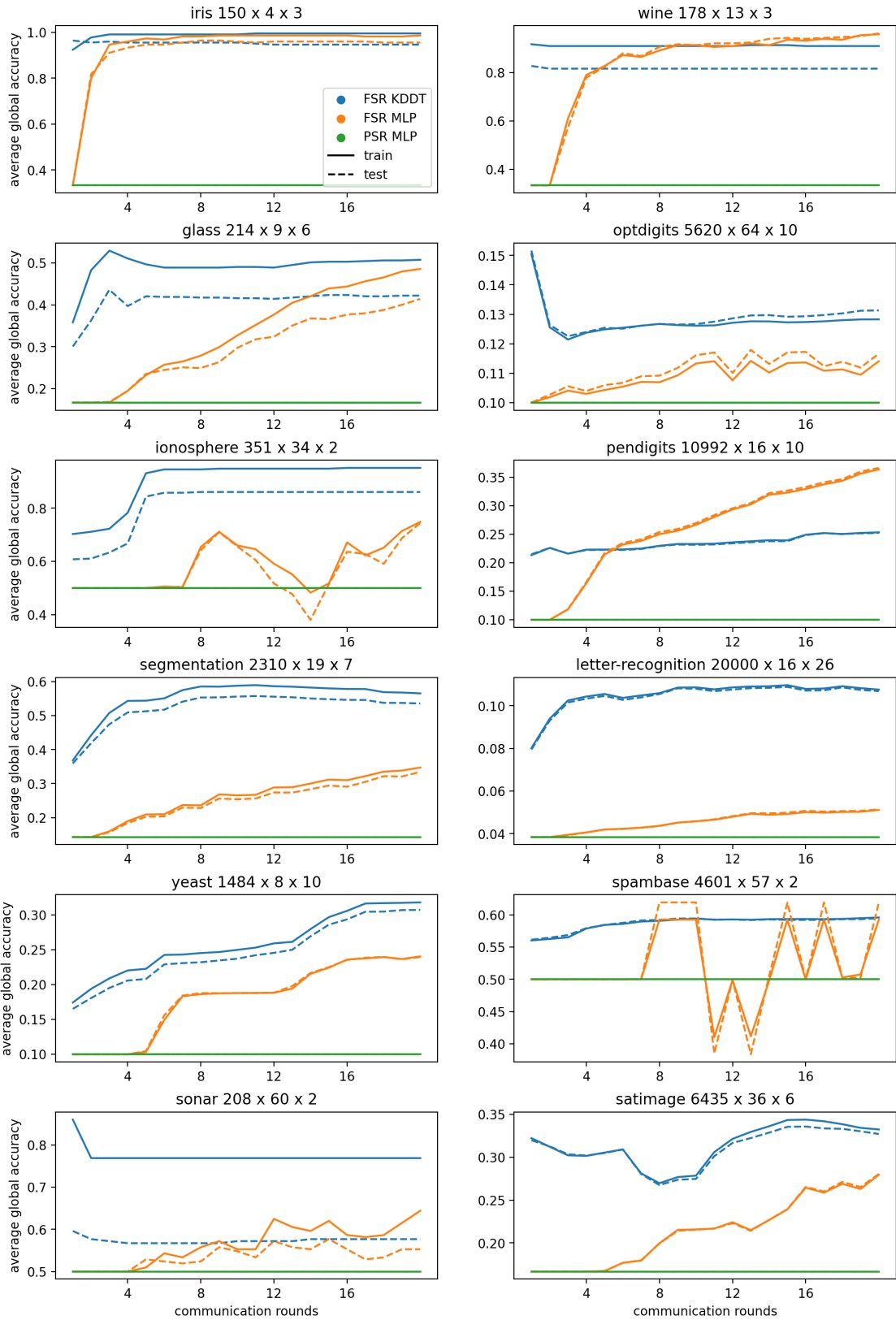

Figure 2: Results for FL experiments with data split by class.

MLP predictably perform differently, with each outperforming the other about half the time. In these experiments, the tree size is not tuned by cross-validation, and the other hyperparameters are selected by training accuracy, so the trees sometimes overfit more than the MLPs.

On a few data sets, the FSR-based methods fall short in performance. The worst cases are optdigits and pendigits, image data where the number of informative features is likely to be high; this is consistent with our expectations for the limitations of the method in its current form, but ongoing efforts show promise towards improving the resilience of the method to higher-dimensional data.

The results for the class-based data split are shown in Figure 1. This is the most extreme possible hetero-geneous split and, unsurprisingly, the PSR method is unable to learn anything in these experiments. For FSR, the results vary, but it is clearly learning in all cases, which is a significant feat for this kind of data split, especially in so few iterations. In most cases, convergence is slower compared to the cluster-based split. This is due in part to the fact that, for data sets with more than two classes, the ring graph implies that it takes longer for information to propagate across all clients. It is also due to to the inherent challenge of the class-based split itself. In simple, low-dimensional examples such as Iris, our methods perform very well, still reaching good performance in few iterations. In the more challenging cases, learning is slower and the accuracy is sometimes unstable across iterations.

It is interesting to note that, despite the simple Monte Carlo method used to apply our FSR method when training the MLP (we use 1000 uniformly random samples for regularization per minibatch, which is very reasonable), stable learning is possible even on moderately-dimensional data where sampling random noise would seem not to cover the domain well. In these cases, the (randomized) error smoothing is crucial: there are cases where, without error smoothing, the disagreement penalty does nothing to improve accuracy compared to local learning alone; however, with it, performance is very good.

## 7 Discussion

This work lays the theoretical foundation and gives an empirical illustration of the proposed method FedFun; below we outline some important limitations and directions for future development.

Not present in this work is a more comprehensive evaluation of FedFun's performance along dimensions such as cost and privacy. While our approach is promising for reducing the total communication iterations needed to learn a consensus model, the need to compute or approximate a function inner product can significantly increase the cost of the local learning problem at each iteration, depending on the model design, inner product, connectivity of the network, and other factors. Moreover, our method, like DJAM, fully solves a local optimization problem at each iteration, and therefore has higher computational cost per iteration compared to methods like DFedAvgM that only perform a fixed number of updates at each iteration. A careful exploration of this tradeoff, between inter- and intra-client costs, would be of interest. Similarly, privacy concerns, i.e. the protection of client data from leakage, are a primary motivation for using FL in many applications. Privacy depends partially on the model, partially on the distributed learning algorithm, and partially on the method of communication. It is left to future investigation to study the privacy implications of the proposed algorithm, as well as the models it introduces as candidates for federated learning, such as decision trees.

Additionally, several developments would enhance usability of FedFun. The model-agnostic Monte Carlo method proposed in Section 5.2 for computing function space inner products and norms is simple to imple-ment, but suffers from poor sample efficiency as the dimensionality of the domain grows large. Exploring inner products and sampling strategies better aligned with risk for different problems and model classes has the potential to improve performance and sample efficiency for high-dimensional data of various modalities. Our algorithm also introduces hyperparameters in the form of the regularization coefficient $\lambda$ and, if using smoothed error, the size and shape of the kernel. While the convergence theory suggests setting $\lambda$ based on $\mu$, $\mu$ may be very large; for instance, with box kernel with radius $\delta$ in $p$ dimensions, $\mu$ is proportional to $(2\delta)^p$. Learning algorithms are not likely to work well with extremely strong penalties like this. Further work is needed to explore how to tune these values, particularly given our distributed setting where computing global performance metrics without exchanging data may not be straightforward.

Lastly, there are various ways in which a network of learning agents may be unstable or evolve over time, ranging from the introduction of new clients and data, to shifting network connections, to the varying availability of clients to participate, to the permanent loss of some clients. For example, Beltrán et al. (2023) highlight military and vehicular applications as a domain where these challenges are especially prevalent. Extensions handling asynchronicity or possible client dropout would increase utility. Many existing works address these challenges for other distributed optimization algorithms, and it is likely that many can be adapted for use with ours.

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

## A   Proofs

This section contains proofs omitted from the main text.

**Lemma 1.** *Let $\varphi$ be a quadratic function $\varphi(h) = \frac{1}{2}\langle h, Ah \rangle + \langle a, h \rangle + \alpha$ on a Hilbert space $\mathcal{H}$ with a minimum value $\varphi^*$. If $A$ is positive with spectral gap $\sigma(A) \cap (0, c) = \emptyset$, then*

$$\|\nabla\varphi[h]\| \geq cd(h, \arg\min \varphi) \tag{6}$$

$$\varphi[h] \geq \varphi^* + \frac{1}{2}cd^2(h, \arg\min \varphi) \tag{7}$$

*for any $h \in \mathcal{H}$.*

*Proof.* Since $a$ and $\alpha$ only shift $\varphi$, we may assume $a = 0$ and $\alpha = 0$ without loss of generality.

By the spectral theorem for bounded operators (Reed & Simon, 1981, Theorem VII.3), $A$ is unitarily equivalent to a multiplication operator: there exists a finite measure space $(\mathcal{X}, \Sigma, \mu)$, a bounded measurable $f : \mathcal{X} \to \mathbb{R}$, and a unitary $U : \mathcal{H} \to L^2(\mathcal{X}, \mu)$ satisfying $U^{-1}T_fU = A$ where $T_f$ is the multiplication operator $[T_f g](x) = f(x)g(x)$. The spectrum of $T_f$ is the essential range of $f$.

We first show (6). For a given $h \in \mathcal{H}$, let $g = Uh$. Denote the projection of $g$ onto the kernel of $T_f$ by $g^* = P_{T_f}g$ with $h^* = U^{-1}g^*$. Denoting the zero set of $f$ by $O_f$, and since $U$ is unitary,

$$\begin{aligned}
\|\nabla\varphi[h]\|^2 &= \|Ah\|^2 \\
&= \|U^{-1}T_fUh\|^2 \\
&= \|T_f g\|^2 \\
&= \int_{\mathcal{X}} |f(x)g(x)|^2 d\mu(x) \\
&= \int_{\mathcal{X}\setminus O_f} |f(x)g(x)|^2 d\mu(x)
\end{aligned}$$

Since unitary equivalence implies equal spectrum, the essential range of $f$ is nonnegative and takes no positive value less than $c$. Thus we have

$$\begin{aligned}
\|\nabla\varphi[h]\|^2 &\geq c^2 \int_{\mathcal{X}\setminus O_f} |g(x)|^2 \, d\mu(x) \\
&= c^2\|g - g^*\|^2 \\
&= c^2\|h - h^*\|^2.
\end{aligned}$$

Moreover, $\nabla\varphi[h^*] = Ah^* = U^{-1}T_fUU^{-1}g^* = U^{-1}T_fg^* = 0$, so $h^* \in \arg\min \varphi$ and (6) follows.

Next we show (7). Define $g$, $g^*$, and $h^*$ as above. Again since $U$ unitary,

$$
\begin{aligned}
\varphi[h] &= \frac{1}{2}\langle h, Ah \rangle \\
&= \frac{1}{2}\langle h, U^{-1}T_f U h \rangle \\
&= \frac{1}{2}\langle Uh, T_f U h \rangle \\
&= \frac{1}{2}\langle g, T_f g \rangle \\
&= \frac{1}{2}\int_{\mathcal{X}} f(x)g(x)\overline{g(x)}\, d\mu(x) \\
&= \frac{1}{2}\int_{\mathcal{X}} f(x)|g(x)|^2 d\mu(x).
\end{aligned}
$$

Proceeding as in the proof of (6),

$$
\varphi[h] \geq \frac{c}{2}\|h - h^*\|^2
$$

and the claim follows since $\varphi^* = 0$ and $h^* \in \arg\min \varphi$. $\qquad\square$

**Theorem 1.** *There exists some $c > 0$ such that $\sigma(\mathbf{A} + \lambda\mathbf{L}) \cap (0, c) = \emptyset$.*

*Proof.* Let $h \perp \ker(\mathbf{A} + \lambda\mathbf{L})$ with $\|h\| = 1$. By Courant-Fisher (Lieb & Loss, 2001, Theorem 12.1), it is enough to show that $h$ satisfies $\langle(\mathbf{A} + \lambda\mathbf{L})h, h\rangle \geq c > 0$. Recall from assumptions 2 and 4 that both $\mathbf{A}$ and $\lambda\mathbf{L}$ have spectral gaps. Letting $P_A, P_L, P_A^\perp, P_L^\perp$ denote the projections onto $\ker \mathbf{A}$, $\ker \lambda\mathbf{L}$, and their orthogonal complements, self-adjointness and spectral gaps imply

$$
\langle(\mathbf{A} + \lambda\mathbf{L})\mathbf{h}, \mathbf{h}\rangle = \tag{17}
$$
$$
\langle\mathbf{A}(P_A\mathbf{h} + P_A^\perp\mathbf{h}), (P_A\mathbf{h} + P_A^\perp\mathbf{h})\rangle + \lambda\langle\mathbf{L}(P_L\mathbf{h} + P_L^\perp\mathbf{h}), (P_L\mathbf{h} + P_L^\perp\mathbf{h})\rangle = \tag{18}
$$
$$
\langle\mathbf{A}P_A^\perp\mathbf{h}, P_A^\perp\mathbf{h}\rangle + \lambda\langle\mathbf{L}P_L^\perp\mathbf{h}, P_L^\perp\mathbf{h}\rangle \geq c_A\|P_A^\perp\mathbf{h}\|^2 + \lambda c_L\|P_L^\perp\mathbf{h}\|^2. \tag{19}
$$

Consider $\overline{\mathcal{H}^n}$ the quotient of $\mathcal{H}^n$ by $\ker(\mathbf{A} + \lambda\mathbf{L})$, with norm $\|\overline{\mathbf{h}}\|_K = \inf_{\mathbf{g} \in \ker(\mathbf{A}+\lambda\mathbf{L})}\|\mathbf{h} - \mathbf{g}\|$ for $\overline{\mathbf{h}} \in \overline{\mathcal{H}^n}$. Recalling that $\mathbf{A}, \lambda\mathbf{L}$ are positive, note that $\ker(\mathbf{A}+\lambda\mathbf{L}) = \ker(\mathbf{A})\cap\ker(\mathbf{L})$. Define $\mu(h) = \left(\frac{1}{n}\sum_{j=1}^n h_j\right) \in \mathcal{H}$ and consider $\mathbf{k}$ in $\mathcal{H}^n$ with all elements equal to $P_0\mu(h)$ for $P_0$ the projection onto $\cap_{j=1}^n\ker(A_j)$. Clearly $\mathbf{k} \in \ker(\mathbf{A})\cap\ker(\mathbf{L})$ so that $\|\overline{\mathbf{h} - \mathbf{k}}\|_K = \|\overline{\mathbf{h}}-\overline{\mathbf{0}}\|_K = 1$. Then by definition $\|\mathbf{h}-\mathbf{k}\|^2 = \sum_{j=1}^n\|h_j - P_0\mu(h)\|^2 \geq 1$, implying $\sum_{j=1}^n\|h_j - P_0\mu(h)\| \geq 1$ by norm equivalence. Thus by the triangle inequality

$$
\left(\sum_{j=1}^n\|h_j - \mu(h)\| + \|\mu(h) - P_0\mu(h)\|\right) = \left(n\|\mu(h) - P_0\mu(h)\| + \sum_{j=1}^n\|h_j - \mu(h)\|\right) \geq 1.
$$

By Lemma 2, there must exist $j$ such that $n^{3/2}\|\mu(h) - P_{A_j}\mu(h)\| + \sum_j\|h_j - \mu(h)\| \geq 1$. Thus, again employing norm equivalence, we have that

$$
n^{3/2}\sum_j\|\mu(h) - P_{A_j}\mu(h)\| + \sum_j\|h_j - \mu(h)\| \geq 1 \implies n^2\|P_A^\perp P_L\mathbf{h}\| + \sqrt{n}\|P_L^\perp\mathbf{h}\| \geq 1.
$$

If $||P_L^\perp \mathbf{h}|| \geq \frac{1}{n}$ then we have that $c_A ||P_A^\perp \mathbf{h}||^2 + \lambda c_L ||P_L^\perp \mathbf{h}||^2 \geq \frac{\lambda c_L}{n^2}$. Otherwise we must have that $||P_A^\perp P_L \mathbf{h}|| \geq \frac{1-\sqrt{n}||P_L^\perp \mathbf{h}||}{n^2} > \frac{1-(1/\sqrt{n})}{n^2}$. Then it follows that

$$||P_A \mathbf{h} - P_L \mathbf{h}||^2 = \sum_{j=1}^n ||P_{A_j} h_j - \mu(h)||^2 = \sum_{j=1}^n ||P_{A_j}(h_j - \mu(h))||^2 + ||P_{A_j}^\perp \mu(h)||^2$$

$$= ||P_A P_L^\perp \mathbf{h}||^2 + ||P_A^\perp P_L \mathbf{h}||^2 \geq \left( \frac{1 - (1/\sqrt{n})}{n^2} \right)^2.$$

Again by the triangle inequality

$$||P_A^\perp \mathbf{h}|| + ||P_L^\perp \mathbf{h}|| \geq ||P_A^\perp \mathbf{h} - P_L^\perp \mathbf{h}|| = ||(\mathbf{h} - P_A^\perp \mathbf{h}) - (\mathbf{h} - P_L^\perp \mathbf{h})|| = ||P_A \mathbf{h} - P_L \mathbf{h}||$$

$$\text{so that } ||P_A^\perp \mathbf{h}|| + ||P_L^\perp \mathbf{h}|| \geq \frac{1 - (1/\sqrt{n})}{n^2}$$

$$\text{and } ||P_A^\perp \mathbf{h}||^2 + ||P_L^\perp \mathbf{h}||^2 \geq \frac{1}{2} \left( ||P_A^\perp \mathbf{h}|| + ||P_L^\perp \mathbf{h}|| \right)^2 \geq \frac{1}{2} \left( \frac{1 - (1/\sqrt{n})}{n^2} \right)^2$$

$$\text{yielding } c_A ||P_A^\perp \mathbf{h}||^2 + \lambda c_L ||P_L^\perp \mathbf{h}||^2 \geq \frac{\min(c_A, \lambda c_L)}{2} \left( \frac{1 - (1/\sqrt{n})}{n^2} \right)^2.$$

$\square$

**Lemma 2.** *Assume $A_i, A_j$ commute for all $i, j$ and let $P_{A_j}$ and $P_A$ denote the projection operators from $\mathcal{H}$ onto $\ker(A_j)$ and $\cap_{j=1}^n \ker(A_j)$ respectively. Then $||f - P_{A_j} f|| < \epsilon$ for all $j$ implies $||f - P_A f|| < \epsilon \sqrt{n}$.*

*Proof.* Since $A_i, A_j$ are self-adjoint and commute we must have that they are simultaneously diagonalizable (Birman & Solomjak, 2012, Theorem 6.5.1) (see also Feldman): There exists a finite measure space $(\mathcal{X}, \Sigma, \mu)$, bounded measurable $a_i$, and unitary $U : \mathcal{H} \to L^2(\mathcal{X}, \mu)$ satisfying $U^{-1} T_{a_j} U = A_i$ for all $j$ where $T_{a_j}$ is the multiplication operator $[T_{a_j} g](x) = a_j(x) g(x)$. Letting $P_{T_{a_j}}$ and $P_T$ denote the projection operators from $L^2(\mathcal{X}, \mu)$ to $\ker(T_{a_j})$ and $\cap_{j=1}^n \ker(T_{a_j})$, and using surjectivity of $U$,

$$||f - P_{A_j} f|| = \inf_{\{g \in \mathcal{H} \mid U^{-1} T_{a_j} U g = 0\}} ||f - g||$$

$$= \inf_{\{h \in L^2(\mathcal{X}, \mu) \mid T_{a_j} h = 0\}} ||f - U^{-1} h||$$

$$= \inf_{\{h \in L^2(\mathcal{X}, \mu) \mid T_{a_j} h = 0\}} ||Uf - h|| = ||Uf - P_{T_{a_j}} Uf||.$$

Similarly, we have $||f - P_A f|| = ||Uf - P_T Uf||$. Thus we see that it is enough to show $h \in L^2(X, \mu)$ satisfy $||h - P_{T_{a_j}} h|| < \epsilon$ implies $||h - P_T h|| < \epsilon \sqrt{n}$.

Consider the zero sets $O_j = \{x \mid a_j(x) = 0\}$. Note that the projections have the effect of zeroing out $h$ on these sets:

$$||h - P_{T_{a_j}} h||^2 = \inf_{\{g \in L^2(\mathcal{X}, \mu) \mid g \cdot a_j = 0\}} ||h - g||^2$$

$$= \inf_{\{g \in L^2(\mathcal{X}, \mu) \mid g \cdot a_j = 0\}} \int_{\mathcal{X}} |h(x) - g(x)|^2 d\mu(x)$$

$$= \inf_{\{g \in L^2(\mathcal{X}, \mu) \mid g \cdot a_j = 0\}} \int_{O_j} |h(x) - g(x)|^2 d\mu(x) + \int_{O_j^c} |h(x) - g(x)|^2 d\mu(x)$$

$$= \int_{O_j^c} |h(x)|^2 d\mu(x).$$

Similarly, we can show $||h - P_T h||^2 = \int_{(\cap_j O_j)^c} |h(x)|^2 d\mu(x) = \int_{\cup_j O_j^c} |h(x)|^2 d\mu(x)$. Thus $||h - P_T h|| < \sqrt{n}||h - P_{T_{a_j}} h||$ follows by induction since

$$\int_{O_i^c \cup O_j^c} |h(x)|^2 d\mu(x) \leq \int_{O_i^c} |h(x)|^2 d\mu(x) + \int_{O_j^c} |h(x)|^2 d\mu(x).$$

$\square$

**Lemma 3.** *For a given $\lambda$, for any $\tilde{\mathbf{h}} \in \tilde{H}$,*

$$\|\tilde{\mathbf{h}} - \bar{\mathbf{h}}\| \leq \frac{1}{\lambda} \frac{\|\mathbf{A}\|}{\nu} \sqrt{\frac{\|\mathbf{A}\|}{\mu}} d(H^*, \arg\min R) \in O(1/\lambda) \tag{20}$$

*where $\bar{\mathbf{h}} = \mathbf{E}\tilde{\mathbf{h}}$ is the projection of $\tilde{\mathbf{h}}$ into consensus.*

*Proof.* Since $\bar{\mathbf{h}}$ is the projection of $\tilde{\mathbf{h}}$ onto the the minimizers of $\mathbf{h} \mapsto \langle \mathbf{h}, \mathbf{L}\mathbf{h} \rangle$, by Lemma 1, we have the following.

$$\|\tilde{\mathbf{h}} - \bar{\mathbf{h}}\| \leq \frac{1}{\nu} \|\mathbf{L}\tilde{\mathbf{h}}\|$$

By optimality of (4), $\nabla R[\tilde{\mathbf{h}}] + \lambda \mathbf{L}\tilde{\mathbf{h}} = 0$.

$$= \frac{1}{\lambda \nu} \|\nabla R[\tilde{\mathbf{h}}]\|$$

Recall that $\nabla R$ is $\|\mathbf{A}\|$-Lipschitz and $\nabla R[\mathbf{h}] = 0$ for $\mathbf{h} \in \arg\min R$.

$$\leq \frac{\|\mathbf{A}\|}{\lambda \nu} d(\tilde{\mathbf{h}}, \arg\min R)$$
$$= \frac{\|\mathbf{A}\|}{\lambda \nu} \sqrt{\frac{2}{\mu} \left( \frac{1}{2} \mu d^2(\tilde{\mathbf{h}}, \arg\min R) \right)}$$

Let $R^* = \min R$ and apply Lemma 1.

$$\leq \frac{\|\mathbf{A}\|}{\lambda \nu} \sqrt{\frac{2}{\mu} (R[\tilde{\mathbf{h}}] - R^*)}$$

Let $\mathbf{h}^* \in H^*$. Since $\langle \mathbf{h}^*, \mathbf{L}\mathbf{h}^* \rangle = 0$ and $\tilde{\mathbf{h}}$ minimizes (4), we have $R[\tilde{\mathbf{h}}] \leq R[\mathbf{h}^*]$.

$$\leq \frac{\|\mathbf{A}\|}{\lambda \nu} \sqrt{\frac{2}{\mu} (R[\mathbf{h}^*] - R^*)}$$

Again apply the $\|\mathbf{A}\|$-Lipschitzness of $\nabla R$.

$$\leq \frac{\|\mathbf{A}\|}{\lambda \nu} \sqrt{\frac{2}{\mu} \left( \frac{1}{2} \|\mathbf{A}\| d^2(\mathbf{h}^*, \arg\min R) \right)}$$

The claim follows by a simple manipulation. $\square$

**Theorem 3.** *For a given $\lambda$, for any $\tilde{\mathbf{h}} \in \tilde{H}$,*

$$d(\tilde{\mathbf{h}}, H^*) \leq \frac{\|\mathbf{A}\|}{\lambda \nu} \sqrt{\frac{\|\mathbf{A}\|}{\mu}} \left( 1 + \frac{n\|\mathbf{A}\|}{\mu} \right) d(H^*, \arg\min R) \in O(1/\lambda). \tag{9}$$

Table 1: Information about data sets.

| full name | short name | labels | features | samples |
|---|---|---|---|---|
| Iris | iris | 3 | 4 | 150 |
| Wine | wine | 3 | 13 | 178 |
| Glass Identification | glass | 6 | 9 | 214 |
| Optical Recognition of Handwritten Digits | optdigits | 10 | 64 | 5620 |
| Ionosphere | ionosphere | 2 | 34 | 351 |
| Pen-Based Recognition of Handwritten Digits | pendigits | 10 | 16 | 10992 |
| Image Segmentation | segmentation | 7 | 19 | 2310 |
| Letter Recognition | letter-recognition | 26 | 16 | 20000 |
| Yeast | yeast | 10 | 8 | 1484 |
| Spambase | spambase | 2 | 57 | 4601 |
| Connectionist Bench (Sonar, Mines vs. Rocks) | sonar | 2 | 60 | 208 |
| Statlog (Landsat Satellite) | satimage | 6 | 36 | 6435 |

*Proof.* Let $\tilde{\mathbf{h}} \in \tilde{H}$, $\bar{\mathbf{h}}$ the projection of $\tilde{\mathbf{h}}$ into consensus, and $\mathbf{h}^*$ the projection of $\bar{\mathbf{h}}$ into $H^*$.

$$\|\tilde{\mathbf{h}} - \mathbf{h}^*\| \le \|\tilde{\mathbf{h}} - \bar{\mathbf{h}}\| + \|\bar{\mathbf{h}} - \mathbf{h}^*\|$$

Since both $\bar{\mathbf{h}}$ and $\mathbf{h}^*$ are in consensus, their elements are equal.

$$= \|\tilde{\mathbf{h}} - \bar{\mathbf{h}}\| + \sqrt{n}\|\bar{h}_0 - h_0^*\|$$

By assumptions 1 and 4 with Lemma 1, since $A_i$ commuting and having spectral gap $\mu$ implies $\sum_i A_i$ has spectral gap $\mu$, the consensus risk functional $\bar{R}[h] = R[(h, \dots, h)] = \sum_i R_i[h]$ is also convex quadratic with minimum growth rate $\mu$ away from its minimizers, of which $h_0^*$ is one.

$$\le \|\tilde{\mathbf{h}} - \bar{\mathbf{h}}\| + \frac{\sqrt{n}}{\mu}\|\nabla\bar{R}[\bar{h}_0]\|$$
$$= \|\tilde{\mathbf{h}} - \bar{\mathbf{h}}\| + \frac{n}{\mu}\|\mathbf{E}\nabla R[\bar{\mathbf{h}}]\|$$

By optimality of (4), $\nabla R[\tilde{\mathbf{h}}] + \lambda\mathbf{L}\tilde{\mathbf{h}} = \mathbf{0}$. Since $L$ is a symmetric graph Laplacian, its rows and columns sum to zero, so $\mathbf{EL} = \mathbf{LE} = \mathbf{0}$. Then $\mathbf{E}(\nabla R[\tilde{\mathbf{h}}] + \lambda\mathbf{L}\tilde{\mathbf{h}}) = \mathbf{E}\nabla R[\tilde{\mathbf{h}}] = \mathbf{0}$.

$$= \|\tilde{\mathbf{h}} - \bar{\mathbf{h}}\| + \frac{n}{\mu}\|\mathbf{E}\nabla R[\bar{\mathbf{h}}] - \mathbf{E}\nabla R[\tilde{\mathbf{h}}]\|$$
$$\le \|\tilde{\mathbf{h}} - \bar{\mathbf{h}}\| + \frac{n}{\mu}\|\nabla R[\bar{\mathbf{h}}] - \nabla R[\tilde{\mathbf{h}}]\|$$
$$\le \|\tilde{\mathbf{h}} - \bar{\mathbf{h}}\| + \frac{n\|\mathbf{A}\|}{\mu}\|\bar{\mathbf{h}} - \tilde{\mathbf{h}}\|$$
$$= \left(1 + \frac{n\|\mathbf{A}\|}{\mu}\right)\|\tilde{\mathbf{h}} - \bar{\mathbf{h}}\|$$

From here the result is proven by application of Theorem 3. $\qquad\square$

# B   Experiment Details

These are additional details for the federated learning experiments covered in Section 6. The datasets used are described in the following table.

The MLP, which is used for both PSR and FSR experiments, consists of two hidden layers of size 50 with ReLU activations. We use batch size 200 and learning rate 0.001 with the Adam optimizer (Kingma &

Ba, 2014). We train for 10000 iterations (batches) locally, then after each round of communication, train for another 1000 iterations with disagreement penalty. For PSR, we use cross-entropy loss and penalize disagreement as the sum of squared difference in parameters and use coefficient $\lambda = 0.1$, which we observe to work well across data sets. For FSR, we use mean squared error loss and penalize disagreement at 1000 inputs sampled uniformly at random from the domain at each batch. Error smoothing is accomplished by, at each batch, adding random noise sampled from the kernel to the training inputs.

The KDDT uses as growth stopping condition a maximum number of leaves equal to the number of training samples summed over clients or 1000, whichever is smaller.

For FSR methods, $\lambda$ and the box kernel radius $\delta$ are chosen to maximize average global training accuracy. For MLPs, we select from $\lambda \in [10, 100, \dots, 10^7]$ and $\delta \in [0.0, 0.01, 0.02, 0.05, 0.1, 0.2, 0.5]$. For KDDTs, we select from $\lambda \in [10, 100, \dots, 10^5]$ and $\delta \in [0.05, 0.1, 0.2, 0.5]$. These values for $\lambda$ may seem large, but the convergence theory suggests that sometimes they should actually be even higher. The best $\lambda$ is often based more on the local learning algorithm than the convergence of the federated optimization.

In all FSR training, we scale the data such that its bounding box, including smoothing, is $[0, 1]^p$. Though we do this up-front for simplicity, it is also straightforward to accomplish this dynamically on a network by communicating data bounding boxes along with models. This scaling is not theoretically necessary, but it makes the choice of hyperparameters more consistent across data sets and prevents the measure of the domain, which scales the disagreement penalty, from taking on extreme values that may be computationally unfavorable.

