# OpenReview forum: "Decentralized Federated Learning with Function Space Regularization"
_TMLR — Rejected by TMLR_

### Review · Reviewer_cWNJ · 2025-06-16

**Summary Of Contributions:**

This paper introduces an new decentralized federated optimization algorithm called FedFun. The main contribution of FedFun is a regularization term, which encourage connect clients to learning similar models and disconnected clients to learn disimilar models. Two clients are connected if they are neighbors on the decentralized federated communication graph. The authors proved that FedFun has a linear convergence rate and the leaned model is close to the optimal solution set. Furthermore, the closeness is poportional to the co-efficient of the regularization term. The new algorithm FedFun is applicable to both neural networks and decision trees.

**Audience:**

Yes

**Claims And Evidence:**

Yes

**Requested Changes:**

1. Provide real-world senarios where the connected clients need similar models. This could better motivate the regularization term. Many prior works assume clients with similar distributions need similar models. The graph connectivity may not imply distribution similarity.

**Strengths And Weaknesses:**

**Strength**
1. This paper is well-executed. The FedFun algorithm and its theoretical gurantee are clear.
2. The function-space regularization is novel. Many prior works focus on parameter-space regularization but regularization in the parameter space is insuffucient for avoiding model divergence among clients. In contrast, regularization in the function-space is more effective [1].
3. The empirical evidence supports the theoretical advantage. Figure 2 in Section 6 shows that the function-space approach learns solutions with good accuracy but the parameter-space baseline does not converge.
4. The method is applicable to both neural networks and decision trees. Decision trees are importance for many analysis tasks.

**Weakness**
1. The Monte Carlo method for estimating the function-space regularization term can be expensive. Applying the regularization term to UCI dataset with less then 100 features requires 1000 samples per minibatch. Given the high-dimensionality of many datasets (e.g., 256x256 in ImageNet), the sampling and forward-pass overhead can grow fast.
2. The function-space regularization term requires tuning its co-efficiency. However, in the theoretical analysis, we know that a larger co-efficiency is always better because the learned solution is closer to the optimal solution set. The theory and practice has a small discrepancy.

Reference

[1] Benjamin, Ari S., David Rolnick, and Konrad Kording. "Measuring and regularizing networks in function space." ICLR 2019.

---

> ### Author Response · Authors · 2025-08-26
>
> Thank you very much for the feedback and taking time to review our work.
>
> We acknowledge that for some high-dimensional data the Monte Carlo method can be too expensive, and scalability improvements are an important direction for future work. We are optimistic that extensions of the approach could employ dimensionality reduction, encoders, or other sampling techniques to reduce cost and improve performance. For instance, jointly learning the underlying data distribution across the federation could reduce practical complexity of the algorithm by allowing reduced sampling effort in regions of lower data density.
>
> We agree that more motivating examples would also be helpful, and we have added these in our method section: One area is healthcare, where clients may aim to jointly learn the effectiveness of certain treatments, but regulatory restrictions limit the clients with whom they can communicate. Similar problems arise in predictive maintenance, for example detecting faults in equipment. In other settings a distributed network of sensors may aim to learn various models e.g. for object detection, but may only be able to communicate with other sensors within a given range.

---

> > ### Comment · Reviewer_cWNJ · 2025-09-11
> > **Reviewer Comment**
> >
> > Thanks for your response. Could the authors explain why connected clients such as (1) healthcare clients with whom they can communicate and (2) equipments need similar models? From my perspective, connected clients may not require similar models unless they have similar data distributions. However, connectivity alone may not guarantee distributional similarity.

---

> > > ### Author Response · Authors · 2025-09-12
> > >
> > > To clarify, it is not our intention to say that connectivity implies distributional similarity. Connectivity of the communication graph just ensures that information can propagate across all clients (otherwise learning would happen independently in each component).
> > >
> > > The relaxation in FedFun has clients minimize local risks with a regularization penalty ensuring resulting models are similar (see also [1],[2]).  If similar models are not desirable, we don't recommend this approach. In healthcare and maintenance settings, similar models are often desirable -- e.g. predicting treatment response from common clinical features.
> > >
> > > [1] Hanzely, Filip, and Peter Richtárik. "Federated learning of a mixture of global and local models." arXiv preprint arXiv:2002.05516 (2020).
> > >
> > > [2] Almeida, Inês, and Joao Xavier. "Djam: Distributed jacobi asynchronous method for learning personal models." IEEE Signal Processing Letters 25.9 (2018): 1389-1392.

---

### Review · Reviewer_iuLZ · 2025-07-26

**Summary Of Contributions:**

This paper introduces function space regularization for solving decentralized federating learning problems. The paper gives a new algorithm that relies on proximal iterations that are solved via local optimization. The proximal iterations solve a relaxed version of the classic joint minimization problem, the relaxed problem has an extra regularization term that forces the functions on different clients to be similar. It's like FedProx [1] but in function space.

[1] Li, Tian, et al. "Federated optimization in heterogeneous networks." Proceedings of Machine learning and systems 2 (2020): 429-450.

**Audience:**

Yes

**Broader Impact Concerns:**

There are no broader impact concerns of mine, though I am a little bit surprised the authors highlighted military applications of this paper as a motivation for handling asynchronous client participation or dropout. I don't think we need to motivate this problem by appealing to military applications, there are many more (less concerning) applications.

**Claims And Evidence:**

No

**Requested Changes:**

- Please clarify the amount of communication needed, per iteration and in total, for applying your method. Compare it with FedAvg or FedProx.
- Please clarify what you mean by robustness to data heterogeneity.
- Can you do a more thorough comparison with the pros and cons against the theoretical guarantees of FedProx [1] or similar parameter-space algorithms?
- Why is it that the performance of parameter space regularization just does not improve with increasing communication rounds in Figure 2? It's just flat, that's very surprising. If it's training, it should decrease.

**Strengths And Weaknesses:**

- (Strength) The paper makes a convincing argument for adopting function space regularization, especially for non-parameteric models.
- (Weakness) I am having a hard time understanding how this method can be communication-efficient at all for neural networks except for very small datasets. In order to apply an update step, client $k$ needs to know how the (average, global) model $h$ predicts each point $x_i$ versus how its local model predicts it ($h_k (x_i)$). This would require communicating a lot of data for each batch. The same applies to other settings (e.g. for decision trees, we need to compute the inner product between the their prediction functions).
- (Weakness) The paper claims this approach is robust to data heterogeneity, but I don't understand what this means. When we use this term in convex optimization, it usually has means that we have some assumption on the variance of the stochastic gradients, and the convergence of the method does not suffer due to this. However, here this is not clear at all. Since we are using local solvers, the stochasticity of the gradients doesn't matter and what remains is just how different the prediction functions are.  What does robustness here signify?
- (Weakness) The paper relies on a reduction from a hard consensus constraint to a relaxed one, but the guarantees for solution-to-relaxed compared to true solution (given by Theorem 2) is derived only for quadratics. For regularization in parameter space, this is not the case, see [2].
- (Weakness) The literature review is very lacking, there are many papers on regularizing models on different clients to be similar in federated learning (like [1, 2]) that aren't cited. There's a rich, existing literature on this in the field of personalized federated learning.


[2] Hanzely, Filip, and Peter Richtárik. "Federated learning of a mixture of global and local models." arXiv preprint arXiv:2002.05516 (2020).

---

> ### Author Response · Authors · 2025-08-26
>
> Thank you very much for the feedback and taking time to review our work.
>
> We are not communicating data as you describe – rather each client communicates their model to neighbors, who can compute or estimate regularizers as described e.g. in sections 5.2 to 5.4. Thus the communication costs are very similar to FedAvg or FedProx (modulo the difference due to operating in a decentralized rather than centralized regime).
>
> The data heterogeneity we explore is similar to many works employing the Dirichlet distribution to control the extent of non-iid data encountered at each client (see, e.g., [1,2]), where the limiting case yields clients holding data from only a single class (increasing differences in local and global prediction functions and related measures of statistical heterogeneity). FedFun can retain its low communication costs and still converge in these settings.
>
> We tried to highlight related work that is most similar to ours or demonstrated a common theme, but you are correct that the related work could be expanded. We have added additional references in personalized federated learning to give readers more context. While there was a reference to FedProx in the introduction, we also now discuss further alongside the other parametric proximal algorithms, and spend more time discussing differences between convergence results.
>
> The convergence guarantees in DJAM [3], the basis of our parameter space regularization, do not apply due to the nonconvexity of the local risks. So we have no guarantee of progress towards a globally optimal hypothesis in our experiments – clients can consistently find a nearby local optimum predicting only their observed class, as seen in Figure 2. Although [3] suggests no need to tune hyperparameters, very high regularization can sometimes help, but can also result in performance oscillating below chance.
>
> [1] Hsu, Tzu-Ming Harry, Hang Qi, and Matthew Brown. "Measuring the effects of non-identical data distribution for federated visual classification." arXiv preprint arXiv:1909.06335 (2019).
>
> [2] Lin, Tao, et al. "Ensemble distillation for robust model fusion in federated learning." Advances in neural information processing systems 33 (2020): 2351-2363.
>
>
> [3] Almeida, Inês, and Joao Xavier. "Djam: Distributed jacobi asynchronous method for learning personal models." IEEE Signal Processing Letters 25.9 (2018): 1389-1392.

---

### Review · Reviewer_hAAw · 2025-08-12

**Summary Of Contributions:**

This paper makes a strong theoretical contribution by enforcing consensus across clients in function space rather than parameter space with theoretical convergence rate and proof-of-concept experimental results. More specifically, the authors propose an iterative process in which clients exchange models with their neighbors and learn a model with function space regularization (FSR), where the motivation for FSR is similar to that in FedProx.

**Audience:**

Yes

**Claims And Evidence:**

Yes

**Requested Changes:**

See above

**Strengths And Weaknesses:**

Strength:
- The authors provide the convergence rate to demonstrate the theoretical performance of the proposed method
- Limitations are clearly stated, e.g., computational cost trade-offs, sampling inefficiency, and the need for hyperparameter tuning, and the privacy analysis
- The proof of concept experimental results are very promising
- The proposed method can also work with non-parametric models

Weakness:
- Limited baseline comparison. The authors only compare the method against PSR. The paper can be stronger if more baselines are considered.
- The experiments are more like proof-of-concept, as the dataset is relatively simple and the model is small. Additionally, Monte Carlo approximation in high dimensional data can be costly and less effective, so it is difficult to judge the practical impact of the proposed approach

Question:
- How are the 'neighbors' defined in the paper?
- Can the authors comment on the statement that 'we use an iterative process initialized by each client minimizing its local risk'? Would this initialization harm the convergence if clients are very heterogeneous?

---

> ### Author Response · Authors · 2025-08-26
>
> Thank you very much for the feedback and taking time to review our work.
>
> We agree that expanding the experiments to include further baselines, FSR implementations, and datasets is an important step for future work. We also acknowledge that for some high-dimensional data the Monte Carlo approximation can be insufficient, but we are optimistic that extensions of the approach could employ dimensionality reduction, encoders, or other sampling techniques to reduce cost and improve performance.
>
> We have updated our method section to clarify that neighbors in our decentralized setting just amount to clients with a line of communication, i.e. there is an edge between them in a graph of the network topology. Thus at every step client i exchanges models with client j if and only if they are neighbors.
>
> The question on initialization is an interesting one – in theory, as long as local subproblems are solved approximately optimally, then the initialization at each client is not important. In practice, we have seen that starting optimization from the previous iterate can improve convergence, both for cluster and class-based splits (and we have added a note in the experiments section to clarify that this is what we do experimentally). We suspect that this is a result of more collocated local optima reducing the accumulation of error.

---

### Decision · Action_Editor_6EsG · 2025-09-26

**Recommendation:** Reject

**Additional Comments:**

The authors may consider a good motivating application, and show experimental results in a realistic situation to improve the paper.

**Audience:**

Yes

**Audience Explanation:**

There is a large audience for federated learning - though the decentralized version may be less popular.

**Claims And Evidence:**

No

**Claims Explanation:**

This paper proposes a distributed learning method where clients can communicate with each other, and the communication network is represented by a graph. The main idea is to use a regularization term in the function space in the optimization process and use a proximal gradient method (proximal gradient methods are already in use in federated learning). The authors show convergence under the assumption on convexity.

The reviews of this paper are mixed, with more substantial negative comments. Most importantly, the regularization simply insists that the connected clients in the communication graph to have similar/same models. Similarity between client distributions and the communication between clients are two different things. The paper claims that the function space regularization gives "theoretical advantage", however then fails to give any example where their particular regularization method will be relevant. Consequentially, in addition, all numerical experiments are simulated, and it is not clear where an advantage is obtained.

There is a further issue with scalability of local optimizations. In addition, the paper claims to provide a "novel framework" of using function space regularization, but it was noted that it is not entirely new, given function space aggregation has been already proposed in federated learning, and function space regularization appears in multiple papers in the neural network literature.

Based on these observations, I am recommending rejection.

**Resubmission Of Major Revision:**

The authors may consider submitting a major revision at a later time.